# Lateral Pterygoid Muscle Alteration in Patients Treated Surgically Due to Mandibular Head Fractures

**DOI:** 10.3390/jcm12144789

**Published:** 2023-07-20

**Authors:** Marcin Kozakiewicz, Paulina Pruszyńska

**Affiliations:** 1Department of Maxillofacial Surgery, Medical University of Lodz, 113 Żeromskiego St., 90-549 Lodz, Poland; marcin.kozakiewicz@umed.lodz.pl; 2Department of Maxillofacial Surgery, University Clinical Hospital Named the Military Academy of Medicine—Central Veterans Hospital, 113 Żeromskiego St., 90-549 Lodz, Poland

**Keywords:** lateral pterygoid muscle, condylar head, mandible condyle, fracture, surgical treatment, open reduction internal fixation, long-term results, computed tomography

## Abstract

The goal of surgical treatment of mandibular head fracture is to restore anatomical relations; however, it also carries other implications. The purpose of this study is to present the alteration in the size of lateral pterygoid muscles after surgical treatment of unilateral mandibular head fractures and the impact of this change on the range of motion of the mandible. Another issue addressed is the persistence of changes in the appearance of the lateral pterygoid muscles after surgical treatment. In this study, 66 patients with unilateral mandibular head fracture were included. An additional 15 patients from this group who were treated only conservatively were separated as an extra reference group to exclude completely the effect of surgery on the appearance of the pterygoid lateral muscle (even on the opposite side to the surgically treated side). The range of mandibular movements was measured at specific time intervals up to 24 months post-operationally. Then, the lengths and widths of the lateral pterygoid muscles on the operated and healthy site were measured in CT images. The results were compared with a control group which consisted of muscles of the intact site (opposite site to the fracture). A significant reduction in the length-to-width ratio of the lateral pterygoid muscle on the operated side by approx. 20% is observed. This suggests that the muscle becomes more spherical, and thus the range of lateral movement to the contralateral side is permanently reduced.

## 1. Introduction

Maxillofacial trauma is still the most common area treated within oral and maxillofacial surgery departments. In Poland, like in other European countries, most of the fractures affect males (from 3.6:1 to 9.4:1, in Poland 5.1:1) with a mean age of 33 years. Among the etiological factors of the injury, there are mainly assaults, traffic accidents, falls, accidents at work, and sports-related injuries. [1]. In the presented study, the factors of greatest significance were falls, with a representation of 42%, then road traffic accidents (39%), and assaults (18%).

According to literature [2], the most common fracture of the condylar process is a basal fracture (54% of condylar fractures), and the second most common fracture is mandibular head fracture (34% of condylar process fractures). Among all of the mandibular head fractures, there is 8% of type A fracture, 34% of type B, and 73% of type C.

The pathomechanism of the injury most often leads to the antero-medial displacement of the proximal fragment of mandibular head. It follows the contracting lateral pterygoid muscle, eventually reaching the infratemporal fossa. Currently, there are two main ways to reach the proximal fragment and its anatomical reduction [3,4,5,6,7,8]: preauricular access and retroauricular access. Both are percutaneous accesses and allow the direct visualization of the displaced fragment to the infratemporal region. Thus, it is possible to move the inner part of the mandibular head (the proximal fragment in diacapitular fracture) by supporting it from the front, i.e., in the pterygoid fovea. In this way, moving the mandibular head achieves its original anatomical position in glenoid fossa and contact with the distal fragment of mandible. It usually takes more than one manual attempt to accomplish a reduction, even though the muscle relaxant application and the caudal ramus of the mandible is pulled down. It appears that this may affect the postoperative appearance of the lateral pterygoid muscle. However, the extent to which its presentation changes is unknown, as the appearance of this muscle has not previously been studied after surgical treatment of mandibular head fracture. And there is an issue of importance, because it is the muscle that causes the mandibular contralaterotrusion, so it is largely responsible for the patient’s functional recovery [9,10].

The aim of this study was to evaluate the appearance of the pterygoid lateral muscle on computed tomography after surgical treatment of mandibular head fracture and its impact on muscle range of motion.

## 2. Materials and Methods

Sixty-six patients with unilateral mandibular head fractures were included in the study (classification according to Neff was applied [11]—there were 5% of mandibular head fracture type A discovered, 21% of type B, and 74% of type C).

Most of the patients reported to the Emergency Department shortly after the injury and were included in the department’s operational schedule as a priority, as soon as possible. Due to their late reporting to the hospital, 8 patients were treated surgically more than 2 weeks after the injury.

The criterion for patient eligibility for this prospective study was the fact that the mandibular head on the opposite side was not injured, because the reference group consisted of the muscles of the healthy side. 

Patients (Table 1) were divided into two groups—51 of them were treated surgically with ORIF (open reduction and internal fixation) and 15 of them conservatively (i.e., closed treatment). Conservative treatment consisted of maxillomandibular ligation for a week, then elastic guide and mouth-opening exercises for another week, followed by 6 weeks of physiotherapy. Qualification for surgical or conservative treatment was not randomized, but dictated by clinical indications. Due to the medical staff special interest in mandibular condyle fractures, indications for surgical treatment were more restrictive than in other centers (shortening of the mandibular ramus of more than 2 mm and rotation of the proximal fragment of more than 30 degrees, versus shortening of the mandibular ramus of more than 5 mm and rotation of more than 50 degrees in other clinics).

The size of lateral pterygoid muscles was then evaluated in spiral computed tomography (CT). A standard protocol in Lodz’s department is to order a CT scan just after a surgery to assess the accuracy of bone fragment fixation, and 3, 6, 12, and 24 months later to assess the progress of a bone consolidation. All computed tomography images were assessed in RadiAnt DICOM viewer (Medixant, Poland, Poznań, www.radiantviewer.com, accessed on 5 July 2023). The examinations were performed at the same window settings for each case: window height of 60 and width of 400 HU, which are the values that can be adopted for viewing soft tissues in CT images. 

The fractured/operated side and the non-fractured/reference side were studied in both groups. The measurements of the greatest length and width of the lateral pterygoid muscle on both fractured and non-fractured sides were taken. Then, the calculation of length-to-width ratio was performed. The reference points (Figure 1) for the measurements were the center of the mandibular pterygoid fovea (pterygoid pit) and the angle between the outer surface of the lateral plate of the pterygoid processus of the sphenoid bone and the posterior wall of the maxilla. In sagittal dimension of the lateral pterygoid muscle, the measurements were made from the central point in the pterygoid depression in an axis directed to the base of the lemma (which may also correspond to the center of the palate). Measurements of the muscle size in CT images were taken at regular time intervals—just after surgery (or during the first visit after fracture in case of conservative treatment), and 3, 6, 12, and 24 months after.

In addition, the focus was on the functional evaluation of surgically treated patients. Measurements of mandibular movements such as maximal interincisal opening, and ipsilateral and contralateral movement were taken. They were made one day after surgery, and then 6 months after and 12 months after, which is a standard protocol for clinic’s postoperative care.

The effect of the time that elapsed between injury and treatment, the effect of the duration of the surgical procedure, as well as age, gender, the cause of the injury, and the material of the screws used for osteosynthesis were also examined. 

Statistical analysis was performed in Statgraphics Centurion 18 (Statgraphics Technologies Inc., The Plains City, VA, USA). A *p* value of less than 0.05 was considered statistically significant. One-way analysis of variance and regression analysis and independence χ^2^ test were done.

## 3. Results

There is no doubt that surgical correction is the only way to restore the anatomical position of the mandibular head; however, the question is if it also improves the condition of the lateral pterygoid muscle, which, at first glance, right after the injury, shows significant asymmetry in terms of its shape (Figure 2).

### 3.1. Measurements

The study began with measurements of the greatest length and width of the lateral pterygoid muscle. The results are presented below (Table 2 and Table 3). The muscle measurements at the immediate post-operative timepost are expected to be affected by soft tissue trauma; therefore, the measurements were repeated in four other timepoints. It is statistically significant that the muscle on the operated side decreases in length over time (*p* < 0.05), even though the swelling has subsided. A significant disproportion in the width of the muscle is also observed. The width of the muscle on the operated side permanently remains greater comparing to the intact side. 

The results above were used to determine the length-to-width ratio of the muscles. It was noted that, during the 00M period (just after surgery or just after the fracture in case of conservative treatment), there is no significant difference in measurements of the muscle length-to-width ratio depending on the method of treatment (conservative or surgical). There is a statistically significant difference in measurements between the healthy (reference) side and the fractured side in both methods (Figure 3, Table 4).

It was noted that, during later phases of the study, for example, during 03M period (3 months after the surgery or after the injury in the case of conservative treatment), a significant difference in measurements of the muscle length-to-width ratio depending on the method of treatment (conservative or surgical) is observed (Figure 4).

### 3.2. Functional Analysis

Surgically treated patients were analyzed in terms of mandible movements. The measurements were taken on three follow-up visits. Considering the maximal interincisal opening, a significant improvement was found over 12 months from an average of 27.59 mm at the beginning to 42.61 mm after one year (Figure 5, Table 5).

As for ipsilateral movement (towards the fracture), at the first phase, it was measured as 6.03 on average; after 6 months, the range of motion increases to 8.46, to achieve 9.73 after 12 months. Since the *p* value is less than 0.05, there is a statistically significant difference amongst this data (Figure 6, Table 6).

When it comes to the function of a single lateral pterygoid muscle, i.e., movement in the opposite direction to the working one, it was observed that, immediately after surgery, the range of contralateral movement is significantly impaired [only 4.6 ± 2.6 mm], *p* < 0.05. The restriction of the range of motion to the healthy side also persists after 6 months [6.3 ± 2.9 mm]. It was noted that, even 12 months after surgical treatment, the patient does not achieve full lateral range of motion (7.3 ± 2.4 mm) in contrast to the full range of mandibular opening movement regained six months earlier [06M MIO = −38 ± 8 mm and 12M MIO = 43 ± 6 mm] (Figure 7, Table 7).

### 3.3. Factors Affecting Long-Term Functional Outcome

The most significant relationship was noted between the range of contralateral motion and the time that passed between the injury and initiating the treatment (*p* < 0.05). It was observed that, the longer the time to treatment, the worse the range of motion to the healthy side was measured (Figure 8), with correlation coefficient = −0.88, R^2^ = 77.64%.

However, factors such as patient gender, reason of the injury, type of fracture, and fixing material were not found to statistically significantly affect functional outcome. It was noticeable, although also statistically insignificant, that the older the patient, the smaller the range of contralateral movement.

## 4. Discussion

In a growing number of centers, mandibular head fractures are treated surgically. The goal of this treatment is to restore anatomical conditions; however, as we notice, it also carries other implications. They are related, for example, with the lateral pterygoid muscle which is inserted into the pterygoid fovea on the mandible. Basing on CT images and measurements of the muscles, permanent morphological alterations in the lateral pterygoid muscle are observed. 

The core of the research is to evaluate the condition of the muscle at the operated site and its evolution over a long period of time. A reference group is needed to understand the significance of this data. The split-mouth method used in the paper may be insufficient for some readers, because the mandible is, however, one bone and both condylar processes act as a functional unit. Thus, simply operating on one condyle causes some degree of dysfunction on the opposite side of the mandible. In order to understand how this degree of dysfunction may be caused by the surgical procedure itself, an extra refereed group was used, i.e., 15 patients with single condylar fractures, but treated only in a closed fashion. There is no impact of surgery in this control. It seems that such a double reference group gives a good frame of reference for the results obtained.

During the study, it was noted that muscle on the operated side becomes permanently wider and shorter, which suggests its more spherical image. Apart from morphological changes, functional changes were also observed. 

According to different studies, it is established that the proper range of MIO (the greatest distance between the incisal edge of the maxillary central incisor to the incisal edge of the mandibular central incisor, when the mouth is opened as wide as possible painlessly [12]) is 44–54 mm, whilst the normal range of lateral movements is 10 mm [13,14]. During the study, it was noted that the range of lateral movement to the contralateral side is permanently reduced. Can we do something to influence the distant functional outcome? It was noted that fixed factors such as patient’s gender or age do not statistically significantly affect the results. The factor that can be changed by the operator such as the fixing material used also does not affect the functional outcome. Fixation is mostly done with two or three screws. It is thought that two-screw fixation may be sufficient for the correct rigidity of the bone union [15]. In some cases, omitting the third screw application would not change the osseous outcome of osteosynthesis, but would reduce the invasiveness observable in the condition of the lateral pterygoid muscle. Shortening the operation time and reducing the number of bores should support the return of activity in the musculoskeltal system.

On the contrary, a significant relationship was noted between the range of contralateral motion and the time from injury to treatment. The earlier the patient visits the ER and the earlier the treatment is initiated, the better the results will be. The distant functional outcome absolutely indicates the great importance of early surgical treatment. Taking into consideration authors’ clinical experiences, it was noted that, at the current stage of development of surgical techniques, late treatment (osteotomies) of the mandibular head unfortunately lead to poor results.

Taking all of the above into account, this question should be reconsidered: what does the surgical treatment offer the patient for the cost of permanent morphological and functional changes in the lateral pterygoid muscle? According to Neff’s 2020 publication [16], surgical treatment may lead to omitting temporomandibular joint dysfunction (TMD), craniomandibular dysfunction (CMD), osteoarthrosis, pain, and, most importantly, ankylosis. The last complication may even lead to the necessity of TMJ replacement. Even though modern rapid TMJ prototyping techniques that are used nowadays make it possible to avoid some complications [17], it is natural that the position of the muscle will change and, therefore, impact the mandible position.

Given the appropriate indications, both treatment options show acceptable results in most cases; however, it is crucial for therapeutic decision making that the long-term results of conservative treatment are much less predictable and may highly affect the health-related quality of life. According to researchers, closed reduction via intermaxillary fixation or solely functional training will not lead to physiological or anatomic recovery and can, at best, offer neuromuscular adaption of the affected joints [18].

In terms of disturbing poor functional results, the reason seems to be that qualification for surgical treatment is not randomized, but dictated by clinical indications (e.g., patient with mandibular head dislocation will never qualify for conservative treatment, which leads to the conclusion that patients treated surgically are those, whose muscles were more affected by the injury from the beginning). As mentioned before, indications for surgical treatment were restrictive (shortening of the mandibular ramus of more than 2 mm and angulation of the proximal fragment of more than 30 degrees), which suggests that patients treated conservatively were the ones with a better initial state of the muscle.

A weakness of the study is the lack of data when it comes to the functional outcome of the patient treated with closed reduction. Following Skroch [19], further investigation concerning the relationship between remodeling and flaring and alterations of the muscle pull would also help us to understand the issue more thoroughly. For the time being, it is not possible to state with the current study results if they could be generalised to other mandibular condyle fractures.

## 5. Conclusions

To conclude, the appearance of the lateral pterygoid muscle is permanently altered after surgical treatment, together with lowering its action. This is an indication to take all efforts to improve the methods of mandible condylar head treatment.

## Figures and Tables

**Figure 1 jcm-12-04789-f001:**
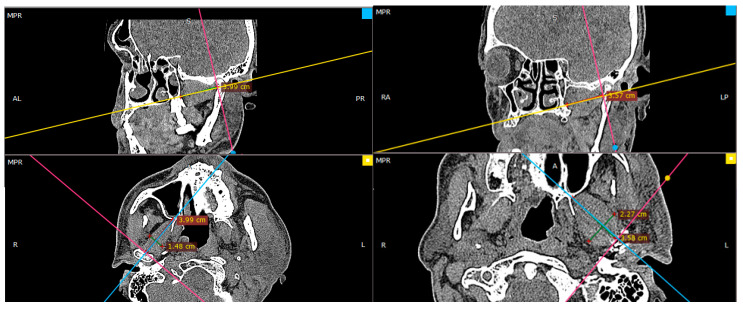
Radiologic imaging of the reference points used for the study—broader description in the text.

**Figure 2 jcm-12-04789-f002:**
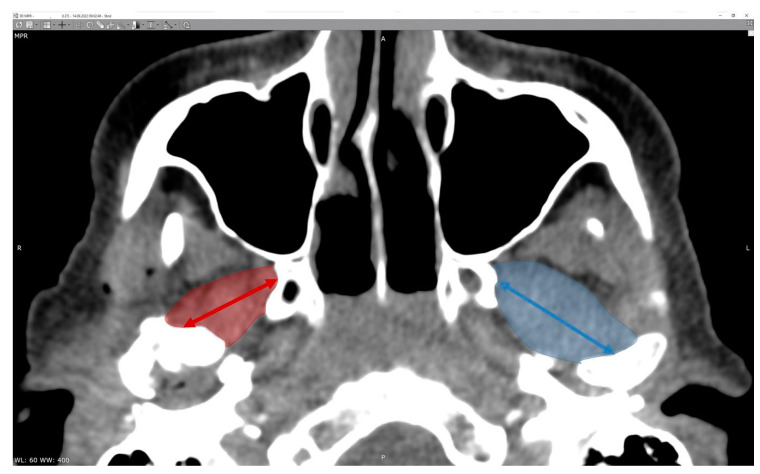
Radiologic imaging study result in a patient with a fracture of the right mandibular head. Shortening of the right lateral pterygoid muscle (red arrow). Please note the length of the muscle on the intact side (blue arrow).

**Figure 3 jcm-12-04789-f003:**
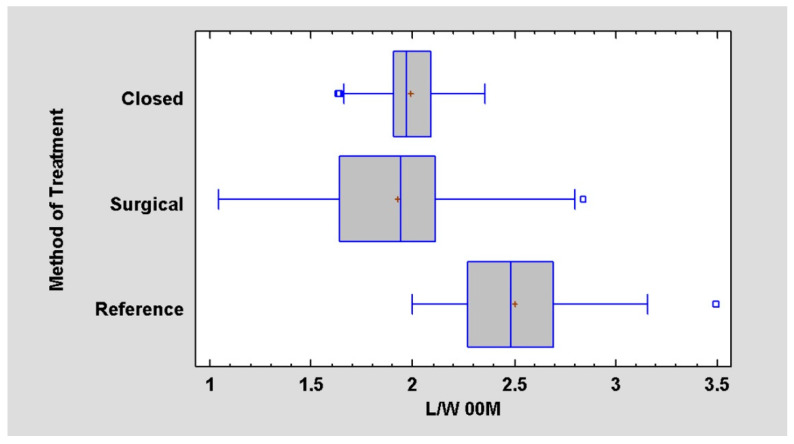
Relationship of the method of treatment to the length-to-width ratio (L/W) at a day of surgery or at a day of injury in case of conservative treatment. Ratio is lower (*p* < 0.05) in affected site (closed/surgical) contrary to intact site (reference site).

**Figure 4 jcm-12-04789-f004:**
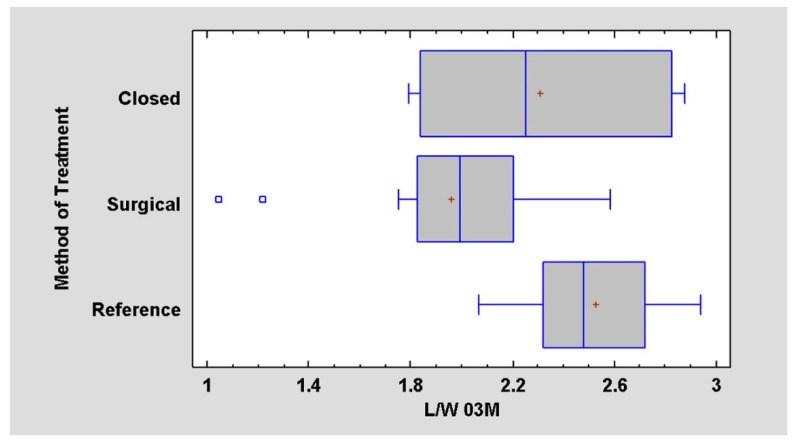
Difference between the treatment outcome in length-to-width ratio at 03M (*p* < 0.05).

**Figure 5 jcm-12-04789-f005:**
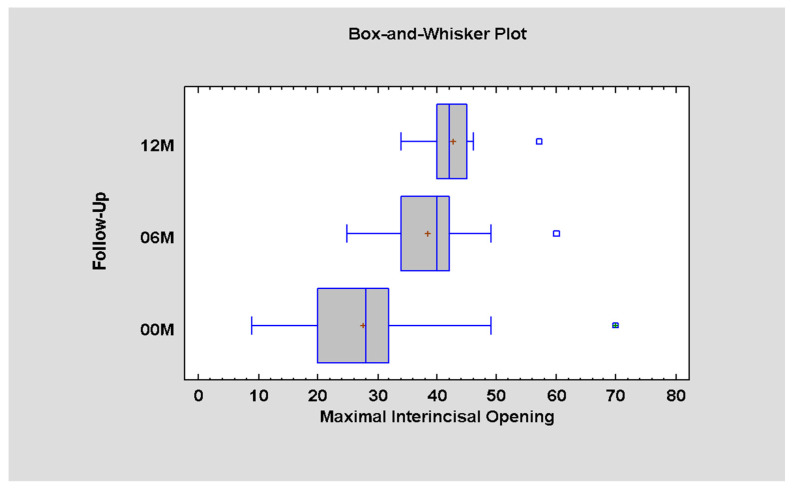
Differences in the MIO measurements and time interval after surgical treatment (*p* < 0.05; there is statistically significant difference between periods 00–06M and 00–12M).

**Figure 6 jcm-12-04789-f006:**
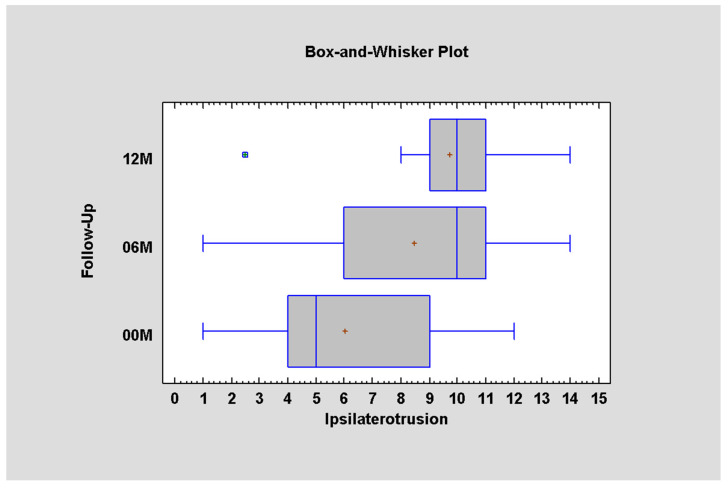
Ipsilateral mandible movement depending on time (*p* < 0.05; there is statistically significant difference between periods 00–06M and 00–12M).

**Figure 7 jcm-12-04789-f007:**
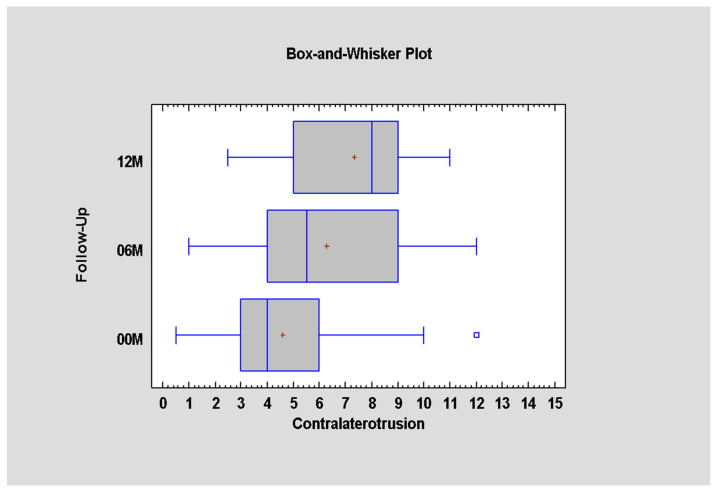
Contralateral mandible movement depending on time (*p* < 0.05; there is statistically significant difference between periods 00–06M and 00–12M).

**Figure 8 jcm-12-04789-f008:**
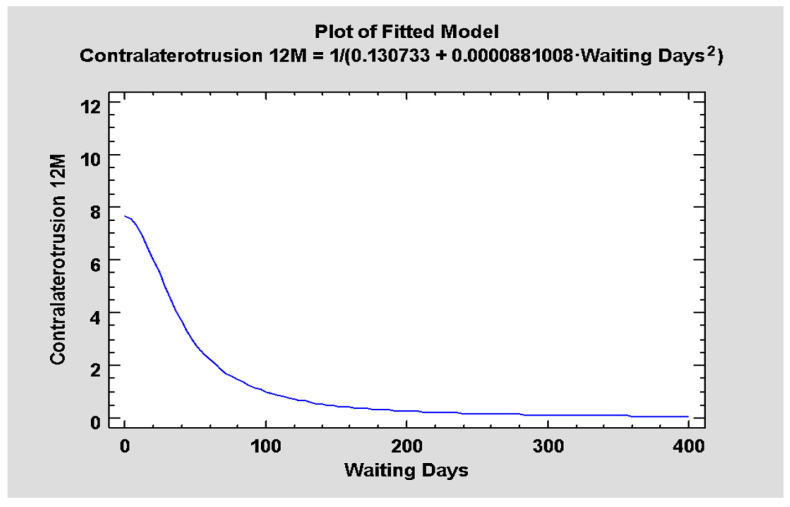
Relationship between the waiting days (delay of surgery) and mandible contralaterotrusion reached 12 post-op (*p* < 0.05).

**Table 1 jcm-12-04789-t001:** Demographic summary characteristic of the study population and its significance regarding the types of mandible condylar head fracture.

	All	Condylar Head Fracture			*p* Value
		Type A	Type B	Type C	
Age	36.8 ± 19.8 years old	34.0 ± 9.0	32.3 ± 20.4	38.3 ± 20.1	0.59 ^2^
Gender	Male:Female = 43:23 ^1^	3:0	9:5	31:18	0.43 ^3^

^1^ significantly younger are females in this traumatologic patients group (*p* < 0.01), i.e., 26.8 ± 14.2 vs. males 42.1 ± 20.4 year old; ^2^ ANOVA; ^3^ test χ^2^.

**Table 2 jcm-12-04789-t002:** Length (cm) of the pterygoid lateral muscle (ANOVA). All presented data are (averages ± standard deviation).

Follow-Up	Intact Side	Surgically Treated	Closed Treatment
00M	3.73 ± 0.32	3.39 ± 0.36	3.07 ± 0.49 ^1^
03M	3.70 ± 0.35	3.23 ± 0.35	3.21 ± 0.74
06M	3.83 ± 0.29	3.27 ± 0.24	3.97 ± 0.01
12M	3.74 ± 0.30	3.08 ± 0.45 ^1^	n/a
24M	3.66 ± 0.36	3.13 ± 0.39 ^1^	n/a
Total	3.73 ± 0.32	3.27 ± 0.38	3.21 ± 0.62
Between-Period Difference	*p* = 0.64	*p* < 0.05	*p* < 0.05

^1^ significantly lower value than in other follow-up periods; n/a—not available.

**Table 3 jcm-12-04789-t003:** Width (cm) of the pterygoid lateral muscle (ANOVA). All presented data are (averages ± standard deviation).

Follow-Up	Intact Side	Surgically Treated	Closed Treatment
00M	1.50 ± 0.16	1.80 ± 0.27 ^1^	1.53 ± 0.09 ^1^
03M	1.47 ± 0.12	1.69 ± 0.34 ^1^	1.38 ± 0.11
06M	1.45 ± 0.15	1.57 ± 0.13	1.44 ± 0.01
12M	1.43 ± 0.15	1.53 ± 0.23	n/a
24M	1.53 ± 0.18	1.60 ± 0.16	n/a
Total	1.50 ± 0.16	1.80 ± 0.27	1.47 ± 0.12
Between-Period Difference	*p* = 0.22	*p* < 0.05	*p* < 0.05

^1^ significantly higher value than in other follow-up periods; n/a—not available.

**Table 4 jcm-12-04789-t004:** Data for length-to-width ratio for 00M.

Method of Treatment	Average ± SD	Minimum	Maximum
Closed	1.99 ± 0.23	1.63	2.35
Surgical	1.92 ± 0.35	1.04	2.84
Reference	2.50 ± 0.29	1.99	3.49

**Table 5 jcm-12-04789-t005:** Data for MIO in millimeters for 00M, 06M, and 12M.

Follow-Up	Average ± SD	Minimum	Maximum
00M	27.59 ± 11.07	9.0	57.0
06M	38.38 ± 7.76	25.0	60.0
12M	42.61 ± 5.77	34.0	70.0

**Table 6 jcm-12-04789-t006:** Data for ipsilaterotrusion in millimeters for 00M, 06M, and 12M.

Follow-Up	Average ± SD	Minimum	Maximum
12M	9.73 ± 2.73	2.5	14.0
06M	8.46 ± 3.47	1.0	14.0
00M	6.03 ± 3.20	1.0	12.0

**Table 7 jcm-12-04789-t007:** Contralaterotrusion immediately post-operation, and 6 and 12 month later.

Follow-Up	Average ± SD	Minimum	Maximum
12M	7.34 ± 2.38	2.5	11.0
06M	6.26 ± 2.92	1.0	12.0
00M	4.60 ± 2.63	0.5	12.0

## Data Availability

The data on which this study is based will be made available upon request at https://www.researchgate.net/profile/Marcin-Kozakiewicz (accessed on 1 June 2023).

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
