# Peer review of "Lateral Pterygoid Muscle Alteration in Patients Treated Surgically Due to Mandibular Head Fractures"

_jcm, 2023, doi:10.3390/jcm12144789_

Round 1

Reviewer 1 Report

The present study analyzes the changes in the lateral pterygoid muscles size after surgical treatment of unilateral mandibular head fractures, compared it to conservative treatment and health control. The measurement were performed on CT images. Furthermore, mandibular movements have been evaluated.

The introduction and the aim is clear even if other details on mandibular head fracture treatment should be developed. Methods and Results are well written. The discussion section is complete. First person plural sentences should be corrected to impersonal form. In my opinion conclusion should be written as a separate section.

Minor editing of English language required

Author Response

Dear Reviewer,

Thank you for spending time on reviewing our manuscript.

According to your note, we corrected the sentences to impersonal form and added a separate section called “Conclusions”.

Reviewer 2 Report

I enjoyed reading your paper titled, “Lateral pterygoid muscle alteration in patients treated surgically due to mandibular head fractures”. You raise an interesting point about functional differences following repair of mandibular fracture and I commend the investigators for considering functional outcomes. The rationale for carrying out the study is well-defined. Some clarification around the purpose of the study will be beneficial in narrowing the focus of comparisons and results throughout the paper. For example, because “dividing 66 patients into two groups who had surgical (n = 51) and conservative (n=15) treatment” in the abstract gave the impression that the primary aim was to compare functional and morphologic outcomes between these two approaches. However, further reading reveals that the focus is not on comparison between surgical and conservative management but rather on measurements between the fractured versus healthy side within individuals. Additional insight into the characteristics of the study population as a whole is warranted, as comparisons based on gender and time to treatment are mentioned, but the manifestation of these variables is not provided to the reader. Revisions around overall clarity and focus are recommended with specific comments presented in the attached document. Of note, my reviews follow the STROBE guidelines for observational studies (can be found here: https://www.equator-network.org/reporting-guidelines/strobe/).

Minor revisions needed in this regard.

Reviewer 3 Report

Manuscript review: jcm-2473114

Congratulations on an interesting study and a refined manuscript.

Title and abstract

- I have no comments on this section.

Introduction

- The beginning of this section should be about epidemiology, mechanism of injury and/or diagnostic methods. I suggest mentioning the osteosynthetic material later in the introduction.

Methodology

- - This section should begin by presenting the design of the study. Was it retrospective or prospective?

- The healthy side of the same patients may not be a valid control sample. Fracture of the mandibular head affects the position of the entire mandible, so the consequence of the injury may also be a change in the dimensions of the lateral pterygoid muscle of the "healthy" side. The hypothesis that, as a consequence of the injury and/or its treatment, the ipsilateral muscle was shortened and the contralateral muscle lengthened should be considered. If you monitored the occlusion throughout the observation, and the control tomographs were made with the teeth connected, it will help to dispel doubts.

- (www.radiantviewer.com access date 25/05/2023) - consider providing the software manufacturer and its registered office instead of this data

Results

- Reconsider that the value should not be negative: "correlation coefficient = 0.88".

Discussion

- I have no comments for this section.

Author Response

Dear Reviewer,

Thank you for spending time on reviewing our manuscript.

As suggested, we added more epidemiological background in the introduction.

Our study was a prospective study, and we also added this information in the text.

The software manufacturer data and correlation coefficiency was corrected according to your suggestions.

The core of the research is to evaluate the condition of the muscle at the operated site and its evolution over a long period of time. A reference group is needed to understand the significance of this data. The split-mouth method used in the paper may be insufficient for some readers, because the mandible is, however, 1 bone and both condylar processes act as a functional unit. Thus, simply operating on one condyle, causes some degree of dysfunction on the opposite side of the mandible. In order to understand what this degree of dysfunction may be caused by the surgical procedure itself, an extra refereed group was used, i.e. 15 patients with single condylar fractures, but treated only in a closed fashion. There is no impact of surgery in this control. It seems that such a double reference group gives a good frame of reference for the results obtained

Round 2

Reviewer 2 Report

Thank you for taking the time necessary on revisions. I have no further comments. Congratulations on a well-written and informative paper.